# Impact Assessment of Free-Roaming Dog Population Management by CNVR in Greater Bangkok

**DOI:** 10.3390/ani13111726

**Published:** 2023-05-23

**Authors:** Elly Hiby, Tuntikorn Rungpatana, Alicja Izydorczyk, Craig Rooney, Mike Harfoot, Robert Christley

**Affiliations:** 1ICAM, Cambridge CB23 7EJ, UK; 2Soi Dog Foundation, Phuket 83110, Thailand; 3Dogs Trust Worldwide, London EC1V 7RQ, UK; 4Vizzuality, Madrid 28010, Spain; 5Dogs Trust, London EC1V 7RQ, UK

**Keywords:** dog, catch, neuter, vaccinate and return, CNVR, dog population management, rabies, dog welfare

## Abstract

**Simple Summary:**

Free-roaming dogs in Greater Bangkok are tolerated and fed by sympathetic citizens. Although many people accept dogs on their street, many more are not accepting of the situation and there are concerns about dog welfare, nuisance behaviors and the ever-present risk of rabies transmission. Catch, Neuter, Vaccinate and Return (CNVR) is an intervention that catches unowned dogs, and collects owned roaming dogs from their owners, for surgical sterilization, vaccination and then return. This prevents these free-roaming dogs from breeding and provides them immunity from rabies. An evaluation of a 5-year CNVR intervention in Greater Bangkok found a reduction in free-roaming dog density, a reduction in dog rabies cases and an improvement in dog-human relationships. Although CNVR was successful at reducing breeding by the current free-roaming dog population, we found evidence that free-roaming dogs are coming from other sources, presumably from abandoned or lost owned dogs. Hence a fully effective dog population management program will require interventions that target abandonment and loss of owned dogs in addition to CNVR.

**Abstract:**

A high-intensity catch, neuter, vaccinate and return (CNVR) intervention was used over 5 years to manage the free-roaming dog population of Greater Bangkok, using nearly 300,000 CNVR operations across six provinces. An evaluation was conducted using multiple methods to assess the impact of this intervention, including clinical data, an observational street survey, an online attitude survey and reported cases of dog rabies confirmed with laboratory testing. The evaluation found evidence of a reduction in free-roaming dog density over time (24.7% reduction over 5 years), a reduction in dog rabies cases (average reduction of 5.7% rabies cases per month) and an improvement in dog–human relationships (a 39% increase per year in free-roaming dogs with visible signs of ownership or care and a perception of less trouble with free-roaming dogs in districts benefiting from CNVR). The CNVR intervention appears to have been effective at reducing the current free-roaming dog population and minimizing one future source of free-roaming dogs by limiting breeding of dogs accessible on the streets. However, there is evidence that other sources of free-roaming dogs exist, presumed to be predominately abandoned or lost owned dogs that were previously inaccessible to the CNVR intervention because they were ordinarily confined or living outside the project area. Hence, fully effective dog population management will require further interventions targeting owned dogs in addition to this CNVR effort.

## 1. Introduction

Domestic dogs have evolved alongside humans for thousands of years to become ubiquitous around the world as both companions and working animals. Current global population estimates stand at between 700 million [1] and 987 million [2] dogs, most of which are free-roaming for at least part of the day. 

Although many dogs are valued by their owners and caretakers, they can have negative consequences for animals, people and the environment, especially when free roaming. These negative consequences include disease transmission, dog bites, road traffic accidents, nuisance behaviours, fouling of public places, and predation of wildlife and livestock. Free-roaming dogs can also experience poor animal welfare. In recognition of these risks, dog population management (DPM) is used to influence dog population dynamics to produce safer populations of dogs in which each dog is wanted, cared for and managed in a way that reduces risks. Dog population management is the subject of international standards [3] and guidelines [4]. 

The interventions required as part of DPM differ according to the local dog population dynamics, how people interact with dogs and the goals of DPM important to the local community [4]. Where unowned dogs are tolerated and fed by local people, often termed ‘community dogs’, catch, neuter, vaccinate and return (CNVR) intervention is appropriate, as it both limits reproduction and builds herd immunity to minimize rabies transmission between dogs and from dogs to people [3]. This approach has been used successfully with community dogs in southeast Asia to control rabies [5,6,7], reduce the density of free-roaming dog populations [6,8], reduce dog bites [9] and improve dog welfare [10,11].

Greater Bangkok is a large metropolitan area with a reportedly high number of free-roaming dogs (for more details see Section 2.1). These dogs appear to be well-tolerated and are commonly fed by sympathetic Bangkok citizens, although there are complaints about free-roaming dogs reported to local authorities and a concern about the risk of rabies transmission from dogs. To achieve the beneficial outcomes of CNVR described previously, a sufficient proportion of the dog population would need to be reached, which can be a challenge across a large metropolitan area [12]. CNVR efforts could be distributed evenly across the metropolitan area or targeted to areas with frequent complaints from residents. However, this is likely to result in low CNVR coverage across the entire area, which may be less effective both in terms of herd immunity for rabies control [13,14] and population decline [15]. Conversely, computer simulation models of animal populations have suggested that high-intensity CNVR is more effective for reducing population size [16,17]. Delivering the majority of CNVR operations in an intensive effort at the beginning of a project leads to greater declines in population size than the same number of operations spread evenly across the project period because the birth of females that would themselves subsequently give birth is prevented by the first intense effort [18]. Soi Dog Foundation is a registered charity with a mission to improve the welfare of dogs and cats in Asia, resulting in better lives for both the animal and human communities. In partnership with Dogs Trust Worldwide, another registered charity with a similar mission to improve dog welfare but with a worldwide remit, Soi Dog Foundation proposed to deliver CNVR using a high-intensity rotational approach, focusing on each district (districts are local authority areas, where the district covers a large area, focus is narrowed further down to the composite subdistricts) until at least 80% of the free-roaming dog population had been reached before moving to the neighbouring district, then, once all districts had been reached, returning to the first district to restart the next ‘round’ of CNVR. 

Evaluations of DPM interventions are essential for adaptive management of interventions and to inform DPM development in other locations but are rarely conducted or published [19], with some notable exceptions [6,8,11,20,21,22,23]. An evaluation of this CNVR approach in Greater Bangkok utilized multiple methods to establish changes in a number of indicators: free-roaming dog density and breeding activity, dog rabies cases, and citizen perceptions and care of free-roaming dogs. 

## 2. Materials and Methods

### 2.1. Study Area

Greater Bangkok, also known as the Bangkok Metropolitan Area, is comprised of the city of Bangkok and five adjacent provinces (Nakhon Pathom, Pathum Thani, Nonthaburi, Samut Prakan and Samut Sakhon). The study area covers nearly 8000 km^2^ and is home to 15 million people living at densities ranging from 700 to 19,500 people per km^2^ [24]. The climate is tropical wet and dry, and land cover is 55% built-up, 33% vegetated, 8% bare and 4% water [25]. According to the last available Bangkok Metropolitan Authority survey, there were 102,490 free-roaming dogs in Greater Bangkok in 2010 [26]. 

### 2.2. CNVR

The CNVR intervention was conducted by mobile teams of dog catchers/handlers and veterinary staff working in field clinics. Field clinics are temporary clinics equipped to conduct surgical sterilisation (including autoclaves, gaseous anaesthesia, preparation and recovery/holding areas) in locations identified as suitable by local community or government leaders, such as community centres or temples. Dog catchers/handlers survey the district or subdistrict in advance to establish the number and locations of free-roaming dogs and to inform and engage the support of dog owners and caretakers/“feeders” in accessing dogs for CNVR (the preclinic survey). Ownership of free-roaming dogs is established by asking households and business in the immediate area about each individual dog. An owned dog is a dog for whom an individual, household or business claims ownership and a right to consent or not to a CNVR operation. A community dog is a dog in whose welfare one or more individuals, households or businesses express an interest and report to provide regular care but do not claim ownership; they may be recruited to help with accessing the dog for CNVR, but this is not a requisite for the definition of a community dog. An unowned dog is a dog for whom no one claims ownership nor interest in providing regular care.

On CNVR operation days, the dog catchers/handlers start in the early morning to catch free-roaming unowned dogs and collect owned and community dogs from owners and community caretakers, including obtaining written consent for surgical sterilisation from owners. These dogs are transported in well-ventilated vans to the field clinics for preoperative examination, surgery and recovery. Once dogs are recovered (usually within the same day but possibly longer depending on the condition of the dog), they are returned as close as possible to the point of capture or directly to their owner. Dogs are marked with an ear tattoo whilst under anaesthesia at the field clinic. This tattoo is a short code devised by Soi Dog Foundation to fit within the ear flap and provide the year, month and location where the dog was caught for CNVR.

Dog catchers/handlers report when the only unsterilised dogs remaining in the area are those that have proven too difficult to catch. They then conduct a visual survey along a convenient and approximately representative route within the area to observe a random sample of the free-roaming dogs to confirm the percentage of dogs that have been caught (the ‘post-clinic survey’). If this percentage is above 80%, the CNVR team moves to the next district or subdistrict. 

The CNVR operation started with just one mobile team in July 2016 and expanded to 6 mobile teams and 2 additional static clinic teams (plus mobile dog catching/handling staff). These teams were staffed by Soi Dog Foundation and a veterinary clinic project partner called ‘Forget Me Not’.

Full standard operating protocols are available in Appendix A. 

### 2.3. Clinical Records

Data for every dog reached by the CVNR intervention were recorded in Microsoft Excel, including their type (unowned, community dog with at least one identified caretaker(s), owned but unconfined for part of the day, or owned and confined to the home), location where they were caught/collected, age (puppy below 6 months or adult), sex, reproductive status (pregnant), weight, medicines received, operation completed, operating veterinarian, any complications and ear tattoo number.

### 2.4. Street Survey

An observational survey of free-roaming dogs was completed in October–November 2016 along 20 routes that covered public roads in Bangkok, Nonthaburi, Samut Prakan and Pathum Thani (the provinces of Nakhon Pathom and Samut Sakhon were not included in this survey but are within the scope of the Greater Bangkok CNVR project). This survey recorded the number of free-roaming dogs seen, their basic demography and their visible welfare state. This survey was replicated in 2020 and 2021, keeping as close as possible to the original survey protocol, to estimate changes in the free-roaming dog population. Data were recorded on paper in 2016 and on the Talea app and web survey tool in 2020 and 2021 (https://www.icam-coalition.org/tool/talea-street-survey-app/, accessed 9 May 2023). The full survey protocol is available in Appendix A. Data were downloaded from the Talea website to Microsoft Excel and analysed using R version 4.2.1, [27] via RStudio [28].

### 2.5. Variable Calculation from Street Survey and Clinical Records

Clinical records were used to calculate two variables for analysis and their impact on dog density: the sterilisation rate and the proportion of spays.

#### 2.5.1. Calculating the Variable ‘Sterilisation Rate’ for Each Survey Route at the Time of Each Survey 

It was assumed that the sterilisation effort at the time of the 2016 survey was 0. For each district, the total number of sterilisations up to the time of the survey event was divided by the total street length within that district. This led to an estimated number of sterilisations per kilometre (km) of street for that district—an estimate of CNVR that took into account the size of the district as represented by street length (free-roaming dogs tend to be associated with streets rather than the space between streets, which is usually filled with buildings; hence, street length rather than area is used as a measure of habitat capacity). 

Nearly all 20 survey routes passed through more than one district, so an estimate of CNVR effort by route required further calculation. For each route, the number of kilometres contributed by each district was calculated. The number of km contributed by each district towards a route was multiplied by the sterilisations per km for that district; the resulting numbers of sterilisations were summed for all districts traversed by that route, then divided by the total route length in km, creating a single ‘weighted mean sterilisation’ for each route at that time; this is the average number of sterilisations performed per km of route, taking into account the street length or ‘weight’ provided by each district. 

The next step was to standardise the weighted mean sterilisation by time to obtain the average sterilisation effort per year for each route; this is the average number of sterilisations performed per km of route per year. The weighted mean sterilisation value calculated from the 2016–2020 sterilisation data were divided by 4 to obtain the average sterilisations per year at the time of the 2020 survey, and the weighted mean sterilisation value calculated from the 2016–2021 sterilisation data were divided by 5 to obtain a measure of sterilisation effort at the time of the 2021 survey. 

The final stage of standardisation involved dividing the weighted mean sterilisation per year for each route by the starting density of dogs on that route in 2016 because some routes had a very low density of dogs in 2016, so the weighted mean sterilisation per year was low, as there were not many dogs available to sterilise. However, this lower number of sterilisations still provided a good sterilisation coverage of the available dogs. The final sterilisation rate is therefore the average number of sterilisations performed per km of route per year as a proportion of the available dogs according to the density recorded in the 2016 survey. Hence, a value of 1 implies that 100% of the available dogs were sterilised per year.

#### 2.5.2. Calculating the Variable ‘Proportion Females’ for Each Survey Route at the Time of Each Survey Event: A Measure of Prioritisation of Females Spays within the CNVR Effort

For each district, the number of females spayed was divided by the total sterilisations recorded up to the time of the survey. This is a measure of the proportion of sterilisations that were female spays for each district in the period preceding the survey.

Nearly all survey routes passed through more than one district, so an estimate of the proportion female spays by route required further calculation. For each route, the number of kms contributed by each district was calculated. The number of kms contributed by each district towards a route was multiplied by the mean sterilisations per km for that district, the resulting number of sterilisations was then multiplied by the proportion of female spays for that district, providing an estimate of the number of females spayed for the section of the route contributed by the district. The estimates of female spays for all contributing districts were summed for the route, then divided by the total number of sterilisations from the contributing sections of the districts. The resulting figure was the mean proportion of females in the sterilisation effort for the route, taking into account the street length or ‘weight’ provided by each district. 

Although the rate of sterilisation and time spent on CNVR could be assumed to be 0 at baseline, it does not make sense to assume a 0 proportion of females at baseline. Hence, the variable proportion of females was calculated using the full 5 years of sterilisation data and used in analysis as a route (higher)-level variable rather than a survey (lower)-level variable. It was also ‘centred’ using a z transformation ((x − mean)/s.d.) so that the mean proportion of females (0.62) became 0 and the values were distributed on either side of 0.

### 2.6. Attitude Survey

An attitude survey was conducted in 2020 to measure public perceptions of free-roaming dogs in Greater Bangkok. This survey included respondents living in areas in which intensive CNVR efforts were conducted and those living in areas of Greater Bangkok that had yet to receive attention from the CNVR project. 

The attitude survey was delivered online because COVID-19 restrictions prevented door-to-door recruitment of respondents in 2020. The attitude survey comprised a consent statement and 13 questions. These questions were translated into Thai and hosted as an online survey using Alchemer (formerly Survey Gizmo), a web-based survey platform. 

It was important that recruitment of respondents was achieved without the respondent knowing that the survey was created by Soi Dog Foundation in order to avoid any potential biasing of responses. Respondents were recruited through two Soi Dog-independent routes: (1) A partnership with the veterinary department of Kasetsart University was established to support recruitment of respondents using a Facebook page that explained the nature of the survey and hosted the link to the survey itself. This Facebook page only displayed the name and logo of Kasetsart University. (2) Contacts within local government offices were asked to distribute the link to the online survey through their resident networks via email without mentioning Soi Dog Foundation. The questionnaire is available in Appendix A.

Data were downloaded from the Alchemer website to Microsoft Excel and analysed using R version 4.2.1, [27] via RStudio [28].

### 2.7. Dog Rabies Cases

The number of government-laboratory-confirmed (99% of samples are confirmed by fluorescent antibody test and 1% confirmed by PCR) animal rabies cases in the Greater Bangkok area from June 2015 to October 2021 was shared with Soi Dog Foundation for analysis of the CNVR effort. These data were stripped of any personal information about each case, leaving only the animal species, ownership status, subdistrict in which the suspect animal was located and the date the sample was submitted for laboratory analysis. 

### 2.8. Statistical Analysis

#### 2.8.1. Analysis of Street Survey Data

##### Free-Roaming Dog Density

A generalised linear mixed model (GLMM) was used with a Poisson distribution for the count of dogs and km of street surveyed as an offset to account for the variation in survey route length. Route was entered as a random intercept to control for the different starting counts of dogs on each route at baseline in 2016. Time was included as a random slope in recognition of differences in how the routes changed in count over time. Time was entered as a fixed effect, calculated as the number of years since the very first survey was conducted on 4 October 2016 as the baseline. The extent of CNVR effort varied across the 20 routes; hence, the interactions between time and two measures of CNVR effort (the sterilisation rate and proportion of spays) were tested for their ability to predict the change in dog density over time by adding these to the GLMM as fixed effects in a three-way interaction with time since baseline. 

##### Breeding Indicators

The street survey provided the measures of two breeding-related demographic variables: the proportion of females that were visibly lactating and the proportion of dogs that were puppies (under 6 months of age). Changes in the proportions of lactating females and puppies over time were analysed using logistic regression, using a GLMM model with binomial distribution, logit link function and route as a random intercept, thereby controlling for differences in starting proportions on routes in 2016. Three variables were tested as main effects and in a 3-way interaction for their ability to predict these breeding-related demographic variables: time since baseline, sterilisation rate and proportion of spays. The relationship between the percentage of lactating females and the percentage of puppies was also analysed using linear regression.

##### Signs of Ownership

The street survey recorded the number of free-roaming dogs with visible signs of ownership or care, including wearing a collar or t-shirt or interacting closely with a person, such as playing or being fed. These signs may not convey legal ownership but do indicate that someone has at least an interest in that individual dog’s fate. A GLMM model with binomial distribution, logit link function and route as a random intercept was used to test three variables as main effects and in a 3-way interaction: time since baseline, sterilisation rate and proportion of spays.

#### 2.8.2. Analysis of Attitude Survey

Binomial logistic regression was used to test the following attitude variables: whether households feed street dogs regularly, a perception of a change in dog numbers over the previous 4 years (reduced versus a combination of the numbers staying the same or increasing), acceptance of dogs on the street (“OK”, “accept” and “happy” combined versus “not accept” and “not accept at all” combined), experience of trouble with free-roaming dogs in the previous month and a perception of trouble experienced today versus 4 years previously (“more trouble 4 years ago” versus a combination of “no change” or “less trouble 4 years ago”). The main predictor tested was where respondents lived, whether in a ‘control’ district where CNVR had not yet been conducted versus a ‘treatment’ district where CNVR had taken place. Whether the respondent owned a dog was also tested as a predictor of whether households regularly fed street dogs. 

#### 2.8.3. Analysis of Dog Rabies Cases

A generalised linear model (GLM) with a negative binomial distribution was used to test whether time since baseline and the number of CNVR operations (sterilistion and vaccination) per month predicted the number of reported dog rabies cases per month. 

### 2.9. Ethical Approval

All procedures performed in studies involving human participants were in accordance with the 1964 Helsinki Declaration and its later amendments or comparable ethical standards. In addition, all methods were approved by the Dogs Trust Ethical Review Board of Dogs Trust UK (reference number ERB037 on 21 October 2020). 

## 3. Results

### 3.1. CNVR Effort

From 1 July 2016 to 30 September 2021, the CNVR project achieved a total of 286,969 sterilisations. These sterilisations were split unequally between the provinces (Table 1), as the CNVR project started in Bangkok city, then expanded to the other five provinces.

The type of dog reached by the CNVR project was recorded for 195,311 (68%) of the dogs, but dog type was not recorded for all dogs early in the CNVR project. The type of dog reached by the CNVR project varied over time (Table 2).

The sterilisation rate for each survey route in 2016 was set at 0; in 2020, the mean was 0.48, with a range spanning a minimum of 0.04 to a maximum of 0.97; in 2021, the mean was 0.47, with a range of 0.15 to 0.90. The proportion of spays was calculated for each route using the full 5 years of data, resulting in a mean of 0.62, a minimum of 0.55 and a maximum of 0.77.

### 3.2. Street Survey

The street surveys included 20 routes totalling approximately 600 km of street and observation of between 1141 and 1626 free-roaming dogs each year (Table 3). Each route was completed within 2–3 h, corresponding to a total of 40–60 h of work for a survey team. In 2016, a single surveyor completed all 20 routes over 20 days; in 2020, four pairs of surveyors completed five routes per pair over 14 days; and in 2021, one lead surveyor completed all 20 routes over 20 days, always accompanied by one of four assistant surveyors, creating a survey pair of recorder and navigator/driver. 

### 3.3. Attitude Survey

The attitude survey collected 757 responses from Greater Bangkok between 30 September and 14 November 2020: a total of 202 from ‘control’ districts where no intensive CNVR effort had yet occurred and 555 from ‘treatment’ districts that had already benefited from CNVR. 

There was a relatively even spread of respondents across the five age categories, the most populous age category was 26–35 years old (25.2%) and the least was 36–45 years old (13.4%), see Appendix A for the breakdown across all age categories. The spread across the five monthly income categories was similarly even, the most populous income category was 15,000–25,000 baht (21.9%) and the least was 7,500–15,000 baht (17.0%), see Appendix A for the breakdown across all income categories. This was reassuring as the online method of recruiting respondents could have created a bias towards younger and more affluent respondents. 

### 3.4. Change in Free-Roaming Dog Density

#### 3.4.1. Change in Free-Roaming Dog Density over Time

The recorded dog density in 2016 was an average of 2.66 (SE 0.29) dogs per km; this fell to 2.47 (SE 0.35) dogs per km in 2020, then to 1.83 (SE 0.28) in 2021, representing an average decline of 0.83 dogs per km (s.d. = 0.19), with the greatest decline of 2.42 dogs per km recorded on route 5 (4.35 dogs per km in 2016 to 1.93 dogs per km in 2021) and the greatest increase of 0.71 dogs per km recorded on route 12 (2.81 dogs per km in 2016 to 3.52 dogs per km in 2021). The GLMM reported a significant reduction in dog density over time (*p* < 0.0001), with an average reduction of 6.5% per year and a 28.5% reduction over the 5 years from 2016 to 2021 (Figure 1).

#### 3.4.2. Impact of CNVR Effort on Change in Free-Roaming Dog Density

The GLMM exploring the impact of the CNVR effort on the change in free-roaming dog density on routes over time reported a significant negative interaction between time, sterilisation rate and the proportion of spays (*p* = 0.001). (Table 4). 

Figure 2 shows the model predictions for the change in the count of free-roaming dogs over time at three levels of proportion of spays and three levels of sterilisation rate (these three levels are the mean, one standard deviation below the mean and one standard deviation above the mean). These graphs indicate that as the sterilisation rate increases, the rate of decline in the count of free-roaming dogs over time also increases. This effect of an increasing rate of decline in free-roaming dogs over time is enhanced when sterilisations are focused on the spaying of females. 

#### 3.4.3. Public Perception of the Change in Free-Roaming Dog Density

The 2020 attitude survey asked residents of Greater Bangkok for their perception of the change in free-roaming dog numbers in the street (“Soi”) on which they lived (Figure 3). The most common perception (31%) of respondents living in ‘treatment’ districts where CNVR had taken place was that there used to be “lots more dogs on my Soi” 4 years ago as compared to today, followed by 28% perceiving “the number to be about the same”, whereas the most common perception (28%) of respondents living in ‘control’ districts where the CNVR project had not yet reached was that there were “far fewer dogs on my Soi” 4 years previously. 

Binomial logistic regression predicted (with significance just above the 5% level) that respondents living in ‘treatment’ districts had odds of reporting a perceived decrease in free-roaming dogs that were 49% higher than those of respondents living in ‘control’ districts (binomial logistic regression: odds ratio, 1.49; 95% CI, 1.00–2.26; *p* = 0.0529). 

### 3.5. Change in Breeding of Free-Roaming Dogs

#### 3.5.1. Lactating Females

Across all routes, the proportion of lactating females decreased from 23.4% in 2016 to 0.9% in 2020 and 1.4% in 2021 (Figure 4). 

Binomial logistic GLMM reported a significant decline over time (*p* < 0.0001), with an odds ratio of 0.45 or a 55% (95% CI: 36–55%) decrease per year in the odds that a female would be lactating. When the sterilisation rate and the proportion of spays were added in a three-way interaction with time, the initial model was overfitted, requiring time as a random slope to be removed for a suitable fit to the data. This model reported a significant negative three-way interaction, indicating that both an increase in sterilisation and an increase in the proportion of spays would lead to a greater decline in the odds that a female would be lactating (Table 5). 

#### 3.5.2. Puppies

Across all routes, the proportion of puppies as compared to adult dogs decreased over time, with 4.8% in 2016, 1.3% in 2020 and 1.1% in 2021 (Figure 5). Binomial logistic GLMM reported a significant decline over time (*p* < 0.0001), with an odds ratio of 0.45 or a 55% (95% CI: 36–55%) decrease per year in the odds that a dog would be a puppy and not an adult. 

The interactions with sterilisation and proportion of spays were found to not predict the proportion of puppies. 

#### 3.5.3. Impact of CNVR on Breeding in Free-Roaming Dogs

Exploring the relationship between lactating females and puppies (under 6 months of age) exposes that the relationship does not follow expectations for a population of free-roaming dogs maintained by only breeding on the streets (Figure 6). We expect the proportion of lactating free-roaming females to predict the proportion of free-roaming pups in a positive direction; more lactating females should be associated with more pups. However, linear regression of 2016 data reports a negative relationship, although this is not significant (linear regression: β = −0.143, R^2^ = 0.135, *p* = 0.11). In 2020 and 2021, there appears to be no relationship at all between the proportion of lactating females and pups (2020 linear regression: β = 0.120, R^2^ = 0.0148, *p* = 0.610; 2021 linear regression: β = 0.044, R^2^ = 0.0145, *p* = 0.613). 

### 3.6. Change in Dog Rabies Cases

There was a total of 234 laboratory-confirmed rabies cases in Greater Bangkok over the 5 years of the CNVR project. There has been a significant decline in rabies cases over time (negative binomial GLM: β = −0.0588, *p* < 0.0001), with an average reduction of 5.7% rabies cases per month (95% CI: 4.6%–6.8%).

The number of CNVR operations (sterilisation and vaccination) per month was found to be a highly significant predictor of the number of rabies cases per month (negative binomial GLM: β = −0.0004401, *p* < 0.0001; Figure 7). An increase in the number of CNVR operations per month by 1000 predicts a decrease of 35.6% in rabies cases per month.

### 3.7. Change in Human–Dog Relationships

#### 3.7.1. Public Perception of Human–Dog Relationships

The 2020 attitude survey asked respondents, “Do you or anyone in your home currently feed any street dogs?”. Overall, 19.8% said they did, with dog owners having over twice the odds of feeding street dogs than non-owners (binomial logistic regression: odds ratio = 2.29, *p* < 0.001). There was no difference between respondents who lived in districts with CNVR and those living in ‘control’ districts that the CNVR project had not yet reached. 

The attitude survey also asked how “accepting” respondents were of the dogs living on their street. Over half of respondents (59%) were not accepting of the dog situation, whilst 40% reported being “OK”, “accepting” or “happy” with the dogs on their street (Figure 8). Responses were combined into two groups: accepting (“OK”, “accept” and “happy”) and not accepting (“not accept” and “not accept at all”). The odds of accepting dogs on their street versus not accepting were not significantly different between respondents living in ‘treatment’ districts and those living in ‘control’ districts (binomial logistic regression, *p* = 0.65).

The attitude survey also asked people whether they had been troubled by free-roaming dogs in the past month; this included many different types of ‘trouble’, such as dog bites, barking/howling and concern for the welfare of free-roaming dogs. A proportion of 40% of respondents reported being troubled by free-roaming dogs in the previous month. The odds of reporting trouble in the previous month were 67% higher for respondents living in ‘control’ districts than respondents living in ‘treatment’ districts (binomial logistic regression: odds ratio = 1.67, *p* = 0.0019; Figure 9).

Respondents were also asked to think back 4 years to 2016 and consider how troubled they were by dogs then as compared to today. The most common response (28%) was that the troubles they have with dogs are about the same, followed by there being more trouble in the past than today (22%) and then less trouble in the past than today (16%) (Figure 10 for responses split by respondent location). Respondents were separated into two groups: those reporting more trouble 4 years ago as compared to today, those reporting a reduction in trouble, and a combination of no change and less trouble in the past as compared to today. Compared to respondents living in ‘control’ districts, those living in ‘treatment’ districts had 45% higher odds of reporting a reduction in trouble over time, although this difference was only significant at the 10% level (binomial logistic regression, odds ratio = 1.45, *p* = 0.10).

#### 3.7.2. Change in Visible Signs of Human–Dog Relationships

The percentage of dogs categorised as showing signs of ownership increased from 32.5% in 2016 to 45.4% in 2020 and 79.8% in 2021 (Figure 11 for route-level results over time). Binomial logistic GLMM reported a highly significant (*p* < 0.0001) effect of time, with an odds ratio of 1.42 or a 42% (95% CI: 31–54%) increase per year in the odds that a dog would show signs of ownership. 

The three-way interaction of time, sterilisation rate and proportion of spays was also statistically significant (*p* = 0.005, Table 6). Figure 12 shows the model predictions for the change in the proportion of dogs showing signs of ownership over time at three levels of proportion of spays and three levels of sterilisation rate (these three levels are the mean, one standard deviation below the mean and one standard deviation above the mean). These graphs indicate that as the sterilisation rate increases, the proportion of dogs with signs of ownership increases, although this effect is modulated by the proportion of spays. At the highest proportion of spays, even low sterilisation rates led to large increases in the proportion of ownership, but with increased rates of sterilisation, the increase in the proportion of ownership does not climb as high as it would with lower proportions of spays. 

## 4. Discussion

The high-intensity rotational CNVR intervention described here involved the sterilisation and vaccination of nearly 300,000 dogs across four provinces of Greater Bangkok over 5 years. Another two provinces have been incorporated into the CNVR project from late 2021 onwards (a small number of CNVR interventions have occurred in these two provinces in response to requests by communities but not at the high intensity achieved by the mobile clinics). Evaluation of this intervention revealed evidence of a reduction in free-roaming dog density over time (24.7% reduction over 5 years), an exponential decline in dog rabies cases (average monthly reduction of 5.7% in rabies cases) and an improvement in dog–human relationships (39% increase per year in free-roaming dogs with visible signs of ownership or care and a perception of less trouble with free-roaming dogs in districts benefiting from CNVR). 

The reduction in dog density was greater on survey routes experiencing higher rates of CNVR effort. This dose-dependent effect is consistent with a causal link between the CNVR intervention and the dog density reduction outcome. Similarly, routes with greater rates of CNVR effort showed greater reductions in lactating females and greater increases in the proportion of dogs with signs of ownership. There was also evidence that increasing sterilisation focus on female spays as opposed to male castrations modulates the effect of CNVR effort and leads to a greater reduction in dog density and lactating females and an increase in the proportion of dogs with signs of ownership. However, findings that combine a high sterilisation rate and a high proportion of spays are an extrapolation from the available dataset. The population of females and males on the street is, on average, close to equal; hence, for the average route, when the sterilisation rate exceeds approximately half of the available dogs, it is not possible to maintain a focus on only female spays. 

These findings suggest that the mechanism for reduction in dog density is a reduction in the breeding capacity of the free-roaming dog population and that females are the limiting factor in breeding capacity. However, the absence of a significant effect of CNVR on the proportion of puppies and the absence of correlation between the proportion of lactating females and proportion of puppies visible during the same street surveys suggests that there was one or more other breeding populations not visible to the survey team. Abandonment of owned dogs inaccessible to the CNVR effort may act as an additional source of free-roaming puppies; these inaccessible dogs include owned confined dogs, dogs living in private gated communities that did not permit access to the CNVR teams and immigrants to the area arriving after the CNVR team had moved on to the next district. 

The reduction in free-roaming dog density appears to have been sufficient to be noticed by local citizens. Attitude survey respondents living in districts that had benefited from CVNR had greater odds of reporting a perceived decrease in the number of dogs on their street over the previous 4 years as compared to those living in districts that had not yet been reached by the CNVR effort. 

In addition to the average reduction of 5.7% in rabies cases per month, there was a significant relationship between the number of CNVR operations per month and the number of dog rabies cases per month. It is assumed that the establishment of herd immunity through vaccination of greater than 70% of the free-roaming dog population predominately led to the reduction in rabies cases [29]. The contribution of sterilisation to rabies reduction could be achieved through a reduction in population turnover, helping to maintain herd immunity within the vaccinated dog population and a reduction in breeding behaviours with associated high contact rates [3,30]. However, as all dogs were both sterilised and vaccinated, the contribution of each element of the CNVR intervention cannot be measured. 

It should be noted that efforts were undertaken by local government to vaccinate owned dogs against rabies in Greater Bangkok in the same time period, which likely also contributed to herd immunity and, therefore, the reduction in dog rabies cases. However, these government campaigns only accessed owned dogs, who may be confined by their owners, whilst the CNVR effort focuses on free-roaming dogs of varying ownership status (owned, community-owned and unowned). As free-roaming dogs have the greatest contact rates with other dogs, they may be defined as ‘epidemiologically relevant’ from the perspective of rabies transmission [31] and are therefore potentially more valuable to herd immunity when vaccinated. 

Many Bangkok residents (20%) provide regular care for free-roaming dogs, and 40% of residents describe themselves as being “OK”, “accepting” and even “happy” about the free-roaming dog situation on their street. However, most residents (59%) report that they do not accept the situation of free-roaming dogs on their street. This combination of findings suggests that although many residents of Bangkok do show compassion towards free-roaming dogs through regular feeding, further improvement in the relationship was desired. A good relationship between people and free-roaming dogs is a prerequisite for CNVR, as an intolerant community does not provide an environment conducive to welfare for a returned sterilised dog. The evaluation revealed two sources of evidence that the relationship between dogs and people has improved over the period of the CNVR project. First, there has been a steep increase (33 to 80%) in the proportion of free-roaming dogs showing visible signs of ownership and/or care. We propose that CNVR is creating greater value in both owned and unowned free-roaming dogs, and people are expressing their connection to these dogs by identifying them in some way to show that they are recognised individuals and have value. Secondly, the attitude survey suggested respondents living in districts that had benefited from CVNR had greater odds of perceiving a reduction in trouble with free-roaming dogs over time and lower odds of having experienced trouble in the previous month as compared to respondents living in districts that had not yet been reached by the CNVR intervention. 

Although we have established evidence of the impact of CNVR on dog density, it only addresses one source of future free-roaming dogs, i.e., breeding by the current free-roaming dog population. Other sources, predominantly abandonment of unwanted owned dogs and loss of owned dogs, are not influenced by CNVR. Evidence that other sources are present in Bangkok include the relatively slow rate of decline in dog density as compared to the steep fall in the proportion of lactating females on the street and the lack of correlation between lactating females and puppies on the street. 

CNVR should be used as part of a wider dog population management system that includes other interventions to address issues such as owned dog acquisition, responsible care of dogs including sterilisation, preventing abandonment and reuniting of lost dogs [4]. ICAM states that the dog population management system must be permanent, but the services within the system, including CNVR, should be adapted over time and appropriate for the local context. In Bangkok, whilst the tolerance and feeding of free-roaming dogs remains high and abandonment continues to provide a significant ongoing source of new free-roaming dogs, CNVR will have an important role to play as part of the DPM system. Removing dogs from the street and housing them in shelters has been proposed as an alternative to CNVR; however, the size of the free-roaming dog population would make sheltering an extremely expensive option as compared to the more cost-effective approach of CNVR [16], and housing in shelters presents significant dog welfare challenges [32].

The repetition of a street survey along 20 routes provided a resource-efficient method of measuring the change in the size of the free-roaming dog population across a wide geographical area [33] corresponding to a total of 40–60 h of work per survey event. The 2021 combination of a lead surveyor to provide consistency in recording and an assistant that was familiar with the local area to provide navigation/driving support appeared to be ideal. The selection of a representative sample of streets for survey routes was conducted separately from the CNVR planning, resulting in routes that traversed areas with a wide variation in CNVR effort. An improvement would have been to design routes that followed the intended rollout of the CNVR effort so that each route was clearly situated within an area of low, medium or high CNVR effort, allowing for clearer identification of the impact of CNVR on dog density changes over time. An additional improvement would have been for surveys between 2016 and 2020 to measure changes in the dog population in the earliest years of CNVR; investment is now being made for a consistent annual survey effort.

The attitude survey was intended to be completed as a face-to-face interview with households ‘at the doorstep’, allowing interviewers to control the sample size coming from districts with and without CNVR intervention and reducing potential biases introduced by an online recruitment method. However, COVID-19 restrictions did not allow for face-to-face interaction, resulting in the decision to move the attitude survey online, with recruitment through Facebook and email distribution lists. Although Internet use is widespread in Greater Bangkok, there is the potential for online recruitment to have introduced bias towards particular subgroups of resident by excluding residents that do not have Internet access and who do not use Facebook or who do not appear in local government email distribution lists. The relatively even distribution of age and income levels indicates that such biases were not obviously along age or affluence lines. To reduce interviewer bias, there was no mention of the Soi Dog Foundation during recruitment of respondents or on the survey form itself. Despite efforts to avoid obvious causes of bias, we cannot be confident that this sample of online respondents is representative of Greater Bangkok citizens.

## 5. Conclusions

High-intensity rotational CNVR in Greater Bangkok has reduced free-roaming dog density, contributed to a reduction in dog rabies cases and led to an improvement in dog–human relationships. CNVR provides management of the current free-roaming dog population in situ on the streets, providing an alternative to the more expensive and less humane option of removal and sheltering of free-roaming dogs. CNVR minimizes breeding in current free-roaming dogs, addressing this source of future free-roaming dogs. However, our evaluation revealed evidence that other sources of free-roaming dogs are relevant, presumably predominately from the owned dog population. Hence, additional interventions focusing on owned dogs will be required in addition to CNVR to create a dog population management system that is effective at addressing all sources of future free-roaming dogs.

## Figures and Tables

**Figure 1 animals-13-01726-f001:**
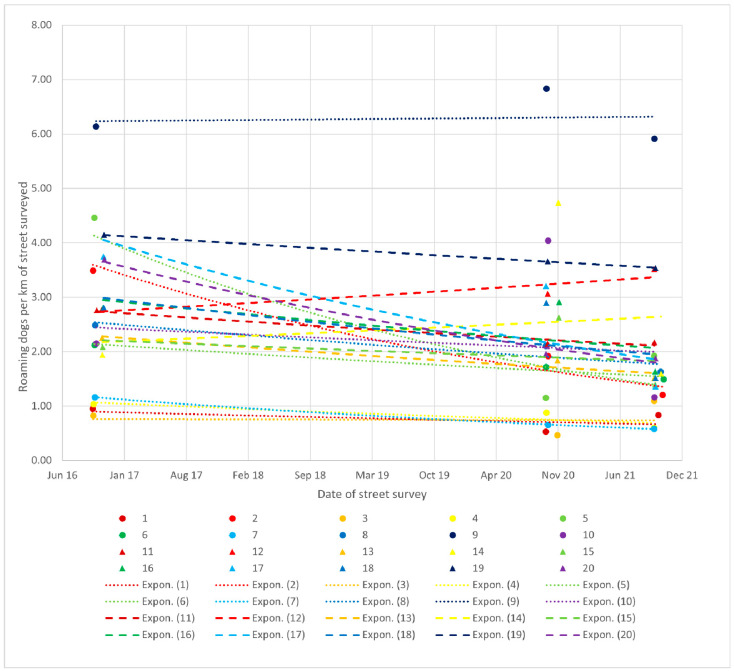
Free-roaming dogs per km of street surveyed along 20 routes in 2016, 2020 and 2021 with exponential trendlines fitted for each route.

**Figure 2 animals-13-01726-f002:**
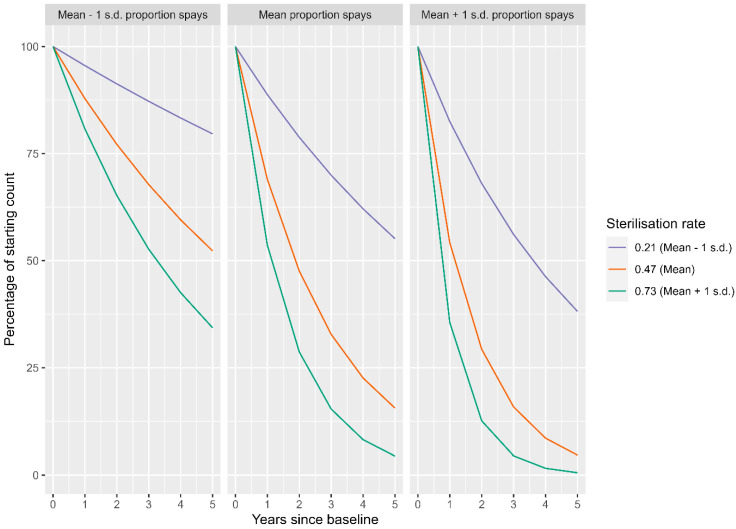
Predicted change in free-roaming dog counts over years as a percentage relative to the starting count in 2016, with varying levels of sterilisation rate and proportion of spays; mean and ±1 standard deviation.

**Figure 3 animals-13-01726-f003:**
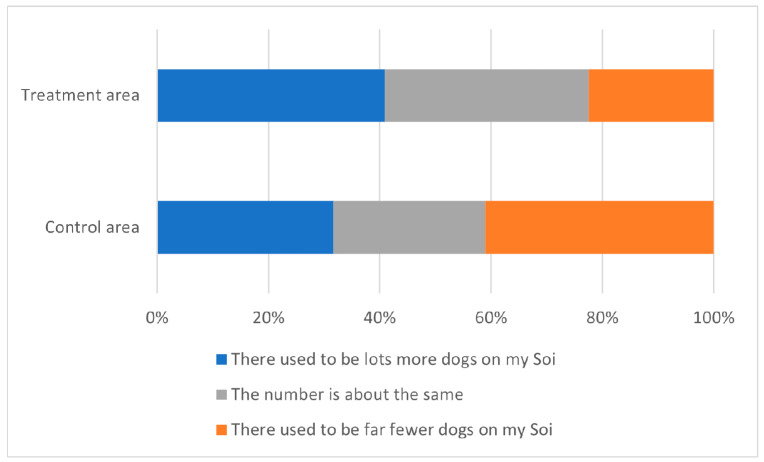
Responses of residents living in ‘treatment’ districts and those living in ‘control’ districts to the question “thinking of the dogs on your Soi today—and then thinking back 4 years ago to 2016—which of the following statements is most true?”. Responses from people who had moved to the area recently or responded “I don’t know” are excluded.

**Figure 4 animals-13-01726-f004:**
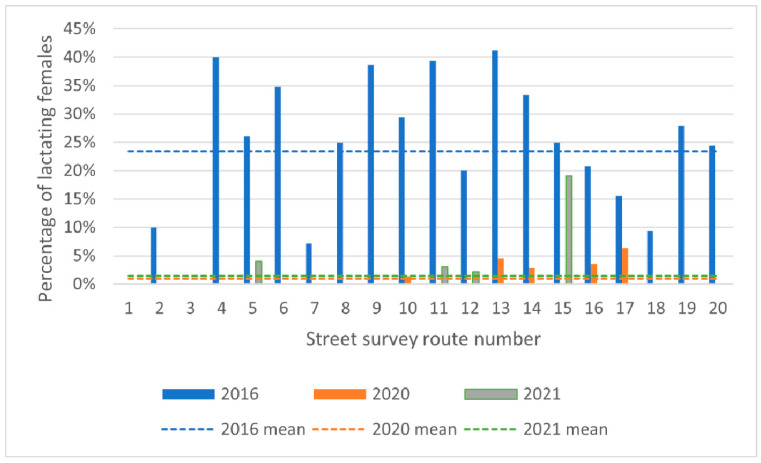
Percentage of females showing signs of lactation in 2016, 2020 and 2021. Dashed lines indicate yearly means.

**Figure 5 animals-13-01726-f005:**
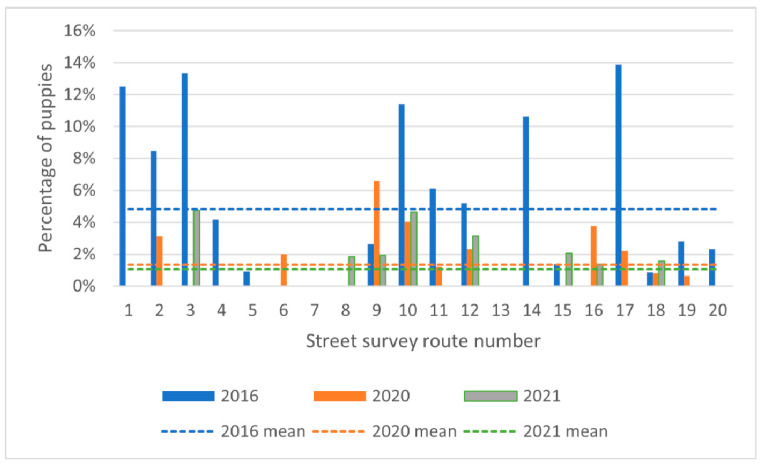
Percentage of observed dogs that are puppies (<6 months of age) in 2016, 2020 and 2021.

**Figure 6 animals-13-01726-f006:**
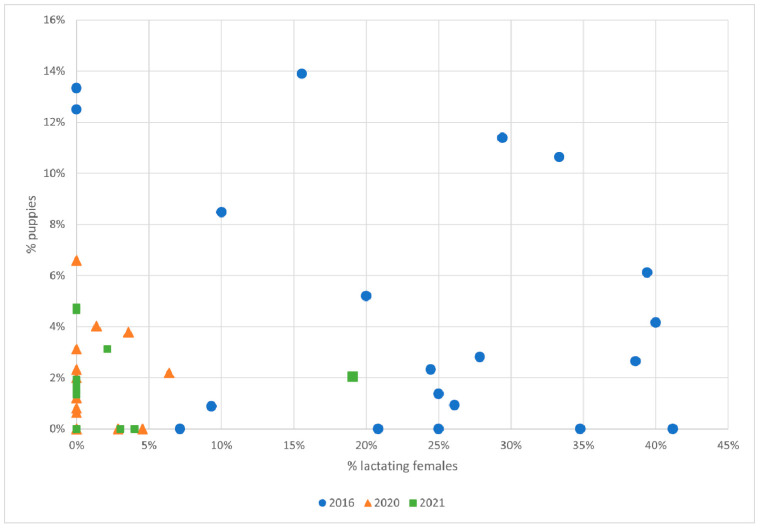
Relationship between the percentage of lactating females and the percentage of puppies in 2016, 2020 and 2021.

**Figure 7 animals-13-01726-f007:**
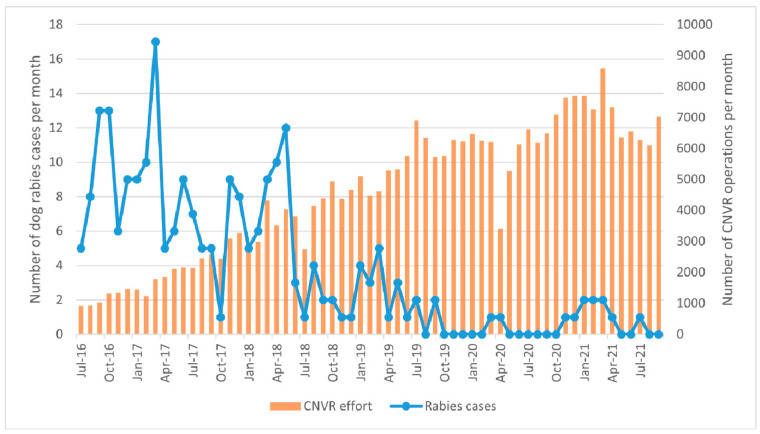
Dog rabies cases and the number of CNVR operations (sterilisation and vaccination) per month over the 5 years of the CNVR project.

**Figure 8 animals-13-01726-f008:**
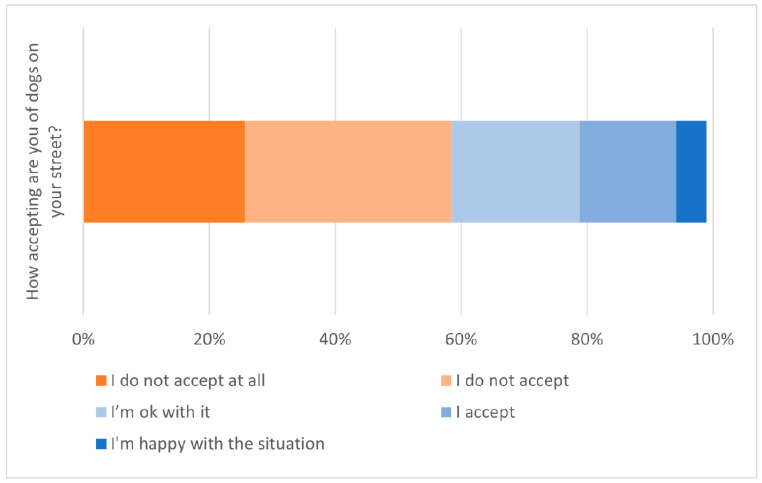
Responses of residents to the question, “How accepting are you of the dogs on your street?”. Responses from people who had moved to the area recently or responded “I don’t know” are excluded.

**Figure 9 animals-13-01726-f009:**
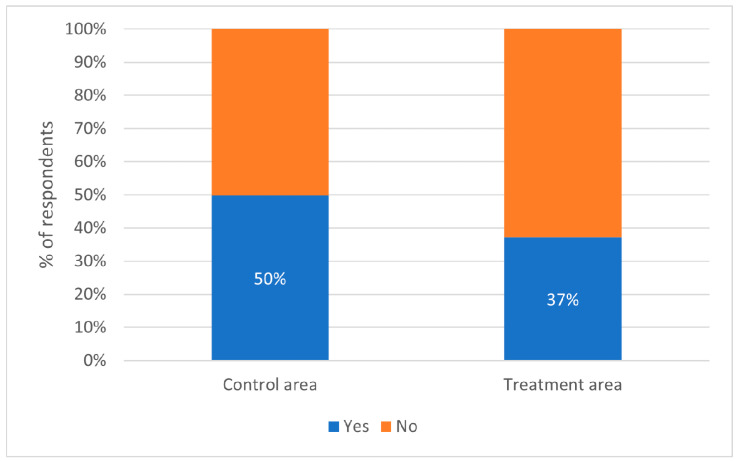
Responses of residents to the question, “In the last month, have you been annoyed or troubled by a dog or dogs in your neighbourhood?” split by those living in ‘control’ districts and those living in ‘treatment’ districts.

**Figure 10 animals-13-01726-f010:**
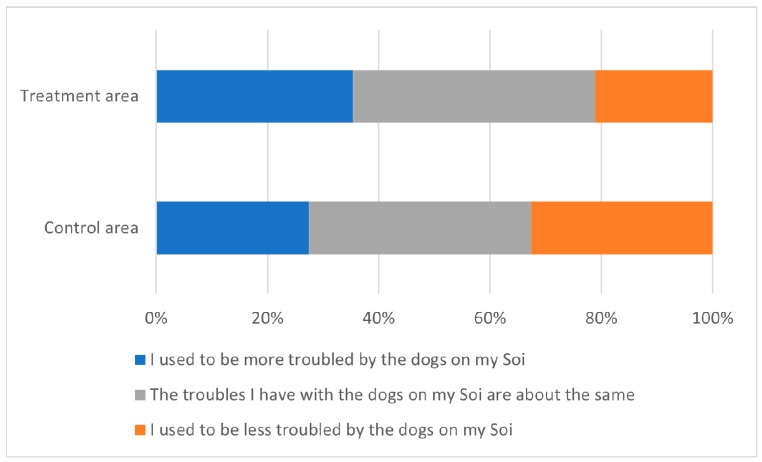
Responses of residents to the question, “Thinking of the dogs on your Soi (street) today—and then thinking back 4 years ago to 2016—which of the following statements is most true?” split by those living in ‘control’ districts and those living in ‘treatment’ districts. Responses from people who had moved to the area recently or responded “I don’t know” are excluded.

**Figure 11 animals-13-01726-f011:**
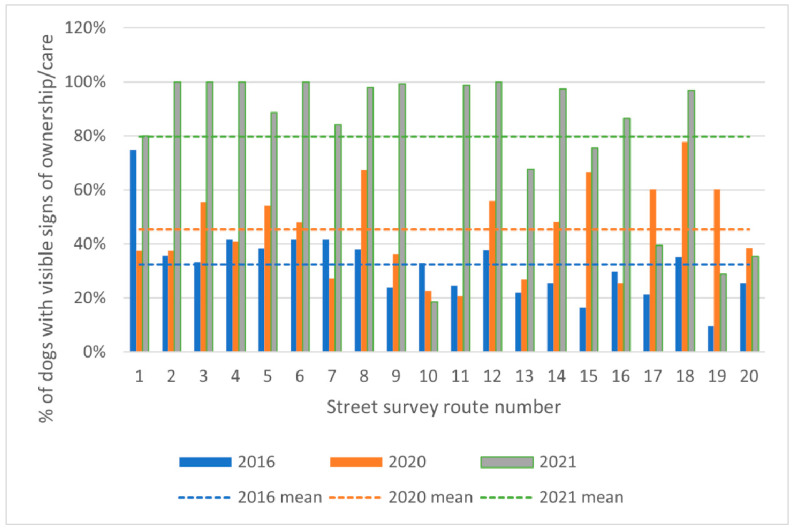
Percentage of dogs with signs of ownership or care recorded on each of the routes used for the street surveys in 2016, 2020 and 2021. Dashed lines indicate yearly means.

**Figure 12 animals-13-01726-f012:**
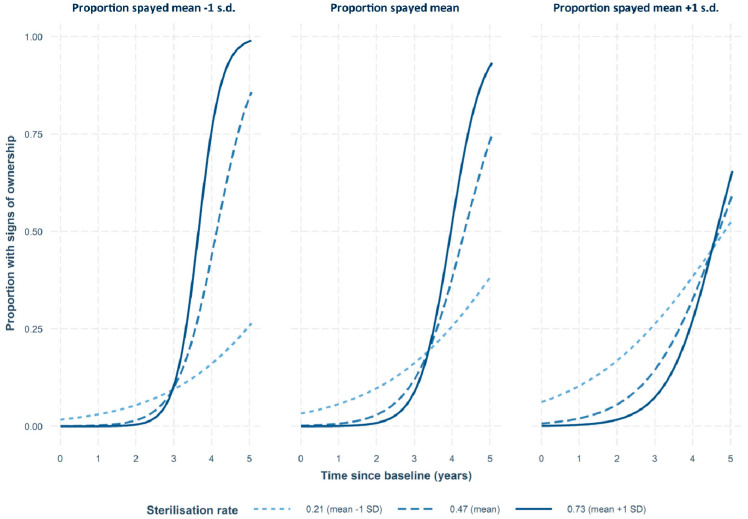
Predicted change in free-roaming dog counts over years as a percentage relative to the starting count in 2016, with varying sterilisation rates and proportions of spays; mean and ±1 standard deviation.

**Table 1 animals-13-01726-t001:** Number of dogs reached by CNVR split by year and by province in Greater Bangkok.

	Bangkok	Nakhon Pathom	Nonthaburi	Pathum Thani	Samut Prakan	Samut Sakhon
2016 (from 1 July)	5736	10	1048	149	34	0
2017	19,509	33	6353	492	172	20
2018	31,449	1155	11,619	1224	779	697
2019	29,027	2585	7287	3796	7011	18,111
2020	23,118	16,562	7520	8747	2086	17,443
2021 (until 30 September)	13,689	22,626	6763	6150	9343	4626
TOTAL	109,352	10,929	31,648	11,848	7222	34,404

**Table 2 animals-13-01726-t002:** CNVR operations by dog type.

Sex		Dog Type			
Female	Male	Unowned	Community	Owned Roaming	Owned Confined
131,198	86,792	45,570	100,300	49,318	123
60%	40%	23%	51%	25%	0%

**Table 3 animals-13-01726-t003:** Summary data for surveys of all 20 routes in 2016, 2020 and 2021.

	Length of Route (km)			Total Dogs Counted			Dogs per km of Street Surveyed		
	2016	2020	2021	2016	2020	2021	2016	2020	2021
Total	596.5	611.7	609.2	1626	1574	1141			
Min	16.8	16.7	16.6	15	9	15	0.82	0.46	0.58
Max	43.9	45.4	45.2	178	182	155	6.08	6.83	5.91
Average	29.8	30.6	30.5	81	79	57	2.66	2.47	1.83
Standard error	1.9	1.9	1.9	10	12	9	0.29	0.35	0.28

**Table 4 animals-13-01726-t004:** GLMM of the effect of time and sterilisation rate on dog density on 20 routes. Significance levels: “.” < 0.10, “**” < 0.001, “***” < 0.0001.

Predictor Variables	Exponentiated Coefficient (Proportional Change in Dog Density)	95% CI	*p* Value	Significance
Years since baseline	1.09	(0.97–1.22)	0.16	
Sterilisation rate	12.55	(4.54–34.66)	<0.0001	***
Proportion of spays	1.08	(0.87–1.34)	0.489	
Years since baseline × sterilisation rate	0.37	(0.27–0.52)	<0.0001	***
Years since baseline × proportion of spays	1.06	(0.97–1.17)	0.211	
Sterilisation rate × proportion spays	3.02	(0.91–10.06)	0.071	.
Years since baseline × sterilisation rate × proportion of spays	0.52	(0.35–0.77)	0.001	**

**Table 5 animals-13-01726-t005:** Logistic regression of the effect of time, sterilisation rate and proportion of spays on lactating females. Significance levels: “*” <0.05, “**” <0.001, “***” <0.0001.

Predictor Variables	Odds Ratio (Lactating vs. Non-Lactating Female)	95% CI	*p* Value	Significance
Years since baseline	0.38	(0.27–0.54)	<0.0001	***
Sterilisation rate	8613	(6.30 × 10^−7^–1.18 × 10^14^)	0.447	
Proportion of spays	0.77	(0.62–0.96)	0.018	*
years since baseline × sterilisation rate	0.12	(4.65 × 10^−4^–29.06)	0.445	
Years since baseline × proportion of spays	1.26	(1.02–1.55)	0.028	*
Sterilisation rate × proportion of spays	1.02 × 10^13^	(1071–9.73 × 10^−22^)	0.0106	*
Years since baseline × sterilisation rate × proportion of spays	0.00062	(3.96 × 10^−6^–0.096)	0.0041	**

**Table 6 animals-13-01726-t006:** Logistic regression of the effect of time, sterilisation rate and the proportion of spays on signs of ownership in free-roaming dogs. Significance levels: “*” <0.05, “**” <0.001, “***” <0.0001.

Predictor Variables	Odds Ratio (Lactating vs. Non-Lactating Female)	95% CI	*p* Value	Significance
Years since baseline	0.85	(0.58–1.24)	0.40	
Sterilisation rate	6.42 × 10^−6^	(5.16 × 10^−7^–7.99 × 10^−5^)	<0.0001	***
Proportion of spays	0.94	(0.77–1.16)	0.60	
Years since baseline × sterilisation rate	34.5	(14.10–84.46)	<0.0001	***
Years since baseline × proportion of spays	1.39	(1.00–1.92)	0.05	*
Sterilisation rate × proportion of spays	31.37	(1.34–733.61)	0.03	*
Years since baseline × sterilisation rate × proportion of spays	0.19	(0.061–0.60)	<0.01	**

## Data Availability

The data presented in this study are available on request from the corresponding author. The data are not publicly available due to the consent agreement with participants.

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
