# Peer review of "Impact Assessment of Free-Roaming Dog Population Management by CNVR in Greater Bangkok"

_animals, 2023, doi:10.3390/ani13111726_

Round 1

Reviewer 1 Report

Comments on the manuscript “Impact assessment of free-roaming dog population management by CNVR in Greater Bangkok” submitted to the Animals

General comments

I appreciate the opportunity to review this exciting manuscript. It brings relevant information on strategies to reduce the occurrence of zoonoses, namely rabies, and increase the well-being of dogs. In this study, the three categories of dogs are actively sought for neutering by the method “Catch, Neuter, Vaccinate and Return” (CNVR). The results are successful because, in five years of the program, there has been a reduction in the number of females in the lactation phase and a general perception of a reduction in the number of dogs on the streets. On the other hand, there has been an increase in puppies. The author raises some hypotheses about the origin of these puppies, which will require a new search and operation strategy in the future.

Street dogs are a cosmopolitan problem (some exceptions exist) and relevant to human and dog health. The description and analysis of the dog population control program has the potential to serve as a reference for programs in many regions of Asia and elsewhere.

The manuscript has an understandable writing, but there are some parts of the text that need some modification. Next, I comment on what seems most appropriate to improve the quality of the manuscript.

Title: Article titles do not have a period.

Keywords: There are too many words. I suggest deleting some words.

Introduction

Line 75: In a nutshell, the reader needs to know more about what Soi Dog Foundation.

Materials and Methods: In this section of the article, the statistical data treatment methods must be presented. The entire description of the data analysis methods, described in the Results section, must be described in the Methods.

Results

Line 245: figures 1 and 2 are redundant (see table 1). Delete figures 1 and 2.

Line 267: Figure 3 is a map that does not have the methodological formality of cartography. Furthermore, it is not relevant to understanding the program described in the manuscript. I strongly suggest deleting figure 3.

Line 301: Figures serve to simplify the visualization of data and analysis. I don't understand the importance of figure 6. Furthermore, it is difficult to interpret due to the overlapping of graphic elements. I question the author if figure 6 is necessary.

Line 400-406: This paragraph is a Discussion.

Line 408: Do not just put “ % “ but write "percentage".

Line 450: In Figure 14, it is not clear what the percentages 100.50% and 205.37% mean.

Author Response

Our thanks to reviewer 1 for their recognition of the value of our manuscript and their constructive feedback. We have made the changes suggested – please see a detailed response below. We did want to note that reviewer 1 discussed “an increase in puppies” – we did not find an increase in puppies, rather a significant decrease over time. But we did find that the % of puppies was not predicted by the % of free-roaming lactating females, suggesting that there was an additional source of puppies on the street.

  • Title:Article titles do not have a period.
    • Agreed, the period has been removed.
  • Keywords: There are too many words. I suggest deleting some words.
    • Agreed, the terms ‘canine’ and ‘impact assessment’ have been removed.
  • Line 75: In a nutshell, the reader needs to know more about what Soi Dog Foundation.
    • We have added a short sentence providing more detail about Soi Dog and their partner, Dogs Trust Worldwide
  • Materials and Methods: In this section of the article, the statistical data treatment methods must be presented. The entire description of the data analysis methods, described in the Results section, must be described in the Methods.
    • Agreed, we have created a new section on statistical analysis and moved the relevant text from the results to this method section.
  • Line 245:figures 1 and 2 are redundant (see table 1). Delete figures 1 and 2.
    • Agreed, deleted
  • Line 267:Figure 3 is a map that does not have the methodological formality of cartography. Furthermore, it is not relevant to understanding the program described in the manuscript. I strongly suggest deleting figure 3.
    • Agreed, moved to supplementary materials where the street survey is described in more detail.
  • Line 301:Figures serve to simplify the visualization of data and analysis. I don't understand the importance of figure 6. Furthermore, it is difficult to interpret due to the overlapping of graphic elements. I question the author if figure 6 is necessary.
    • We have removed the route numbers and leader lines from the graph to make it clearer, but prefer to keep this figure in the manuscript – we agree that it is not as high in importance as other figures, but we feel this visualisation would still be of interest to the reader.
  • Line 400-406: This paragraph is a Discussion.
    • Agreed, this has been removed and the subject has already been sufficiently covered in the discussion.
  • Line 408: Do not just put “ % “ but write "percentage".
    • Corrected as suggested.
  • Line 450:In Figure 14, it is not clear what the percentages 100.50% and 205.37% mean.
    • 100 and 205 were the number of respondents that responded yes from each area, whilst the percentages were 50 and 37. As this was unclear these have been removed from the graph leaving only the percentages.

Reviewer 2 Report

General comments

Dear authors, the paper is very good and presented in a good way. The subject is of interest and the work behind the paper has been huge.

The paper is very long, but actually there is no room for reduction. The long parts are M&M and Results, not Intro nor Discussion

I found that the self-citations are 25% of the total citations. I recognize the importance of ICAM member in the DPM globally, but maybe self-citations can be reduced a bit? 

I know that the paper is big and with many different parts, but sometimes I had the feeling that some parts should be moved in the appropriate section (e.g. things discussed in discussion and not in results). I put some comments in Specific comments below, but I suggest also the authors to self-check 

Specific comments:

L14: across over instead of across cover?

L21: replace "managing" with "reducing"? With "managing" it sounds as the problem is solved

L57: km2 should be superscript. Then kms is used. Be consistent throughout the whole manuscript

L57: reference for the 15 million people would be good

L58: I know links are not permanent, but why don't you provide the actual link here?

L71-75: maybe it's me, but it is not clear 

L90: I suggest you (but it is just a suggestion among peers) to put a short subchapter "study area" with the population data (human and dog) and why don't you use also a CLC or other detail to provide readers with an idea on how is the environment? E.g. How much is urban/rural?

L113: a tattoo saying what? An official ID for Bangkok or Thai Dog Registry or a private company ID?

L127 and Table 2: I would avoid the term stray and use free-roaming. Throught the whole manuscript

L126-143 and other parts: call Excel always in the same way

L216-222: it is not explained here (and it should) that was by email. Only in the discussion reader 

 L226-230: not clear what a rabies case is and it should be defined. Serological, PCR, dead animals examination, observation after bite?

L226: Official? Private? Both?

Fig 1: "2016" is missing, at least in my view

Fig 2: just a suggestion: why don't you use the same size (height is different) of Fig 1?

L259: missing a space between 600 and km

Fig 3: missing scale and legend

L291-297: this is M&M, not Results. In M&M there is no mention of any GLMM (also for other variables) and they must be there 

L400-406: this is discussion, not results

Fig captions: maybe you can remove some text and explain in the text control and treatment meanings?

Fig 14: some problems with percentages (exceeding 100%) or not clear to me?

L472-475: this is M&M instead of Results?

L496-501: this is Discussion

L522: better avoid "casual", the statitical tests used study the association rather than a cause-effect relationship

L562: nd?

L587: t instead of T

Author Response

We are very grateful that Reviewer 2 can see the value of our manuscript and recognises the work that has gone into creating this impact on the dog population in Bangkok. We also wanted to thank them for accepting the length of our manuscript as unavoidable, we agree that it is very long!

Your comments about moving discussion type text from results to the discussion section is well taken, we hope we have achieved that sufficiently in the updated manuscript.

Citations for ICAM and Hiby

  • We reviewed the 4 citations of ICAM 2019 and the 3 citations for different publications by Hiby and others, but could not find alternative references that fully supported the points being made. If the reviewer knows of other alternative references that we have missed, we would welcome these suggestions.

L14: across over instead of across cover?

  • This should just say “across 6 Provinces”, it has been amended by deleting ‘cover’.

L21: replace "managing" with "reducing"? With "managing" it sounds as the problem is solved

  • Agreed, has been changed as suggested.

L57: km2 should be superscript. Then kms is used. Be consistent throughout the whole manuscript.

  • Agreed, 2 has been changed to superscript. And the 2 instances of “kms” are now “km” and the acronym introduced when first used.

L57: reference for the 15 million people would be good

  • Reference included for 2010 census which is a slightly lower number at 14,626,225, rounding to 15 million, the “over” 15 million is a projection of the more recently population size that is no longer visible on the National Statistics Office of Thailand webpage.

L58: I know links are not permanent, but why don't you provide the actual link here?

  • The website address is partially in Thai – we will include it in the reference list and will check whether Animals can publish Thai script.

L71-75: maybe it's me, but it is not clear 

  • This has been reworded

L90: I suggest you (but it is just a suggestion among peers) to put a short subchapter "study area" with the population data (human and dog) and why don't you use also a CLC or other detail to provide readers with an idea on how is the environment? E.g. How much is urban/rural?

  • This was a great suggestion, we have done as you suggested.

L113: a tattoo saying what? An official ID for Bangkok or Thai Dog Registry or a private company ID?

  • The tattoo provides the year, month and a code for the province where the dog was caught – this is a short coding system devised by Soi Dog to fit inside the ear flap and help identify an individual if it was caught again (e.g. for veterinary treatment) - this has been included in the text.

L127 and Table 2: I would avoid the term stray and use free-roaming. Throught the whole manuscript

  • Agreed, the word stray has been removed as it is unnecessary in these 2 places; the key word is “unowned”, all the dog types mentioned are free-roaming aka stray. We have used free-roaming throughout the rest of the manuscript.

L126-143 and other parts: call Excel always in the same way

  • Agreed, amended to Microsoft Excel in all instances.

L216-222: it is not explained here (and it should) that was by email. Only in the discussion reader 

  • We have clarified that distribution was via email in the methods.

 L226-230: not clear what a rabies case is and it should be defined. Serological, PCR, dead animals examination, observation after bite?

  • This has been clarified in the text as fluorescent antibody testing.

L226: Official? Private? Both?

  • Confirmation was done by government laboratories.

Fig 1: "2016" is missing, at least in my view

  • Review 1 has requested that we remove this figure because it is unnecessary in their view.

Fig 2: just a suggestion: why don't you use the same size (height is different) of Fig 1?

  • Review 1 has requested that we remove this figure because it is unnecessary in their view.

L259: missing a space between 600 and km

  • Corrected

Fig 3: missing scale and legend

  • Review 1 has requested that we remove this figure because it is unnecessary in their view.

L291-297: this is M&M, not Results. In M&M there is no mention of any GLMM (also for other variables) and they must be there 

  • Agreed, we have created a new section in the methods on statistical analysis and moved all the relevant text from the results to this method section.

L400-406: this is discussion, not results

  • Agreed, this has been removed and the subject has already been sufficiently covered in the discussion.

Fig captions: maybe you can remove some text and explain in the text control and treatment meanings?

  • Agreed, the explanation of what a control and treatment district are has been moved to the statistical analysis section and removed from the captions.

Fig 14: some problems with percentages (exceeding 100%) or not clear to me?

  • 100 and 205 were the number of respondents that responded yes from each area, whilst the percentages were 50 and 37. As this was unclear, these have been removed from the graph leaving only the percentages.

L472-475: this is M&M instead of Results?

  • Agreed, this has been moved to the new statistical analysis section of the methods.

L496-501: this is Discussion

  • Agreed, this has been moved to the discussion and shortened.

L522: better avoid "casual", the statitical tests used study the association rather than a cause-effect relationship

  • We have purposely gone beyond just association here, we are proposing that because the change in dog density dependent on the ‘dose’ of CNVR this is evidence for a cause-effect relationship and not only an association where the direction of influence in unknown. However, we appreciate that “evidence of” is too strong, so have suggested a change to “is consistent with”.

L562: nd?

  • This book has now been published and will be launched in July 2023, hence nd removed and replaced with 2023.

L587: t instead of T

  • Corrected